# Concentration inequalities under sub-Gaussian and sub-exponential conditions

**Andreas Maurer**
Istituto Italiano di Tecnologia
am@andreas-maurer.eu

**Massimiliano Pontil**
Istituto Italiano di Tecnologia & University College London
massimiliano.pontil@iit.it

## Abstract

We prove analogues of the popular bounded difference inequality (also called McDiarmid's inequality) for functions of independent random variables under sub-Gaussian and sub-exponential conditions. Applied to vector-valued concentration and the method of Rademacher complexities these inequalities allow an easy extension of uniform convergence results for PCA and linear regression to the case of potentially unbounded input- and output variables.

## 1 Introduction

The popular bounded difference inequality [11] has become a standard tool in the analysis of algorithms. It bounds the deviation probability of a function of independent random variables from its mean in terms of the sum of conditional ranges, and may not be applied when these ranges are infinite. This hampers the utility of the inequality in certain situations. It may happen that the conditional ranges are infinite, but that the conditional versions, the random variables obtained by fixing all but one of the arguments of the function, have light tails with exponential decay. In this case we might still expect exponential concentration, but the bounded difference inequality is of no help.

Vershynin's book [14] gives general Hoeffding and Bernstein-type inequalities for sums of independent sub-Gaussian or sub-exponential random variables. In situations where the bounded difference inequality is used, one would like to have analogous bounds for general functions. In this work we use the entropy method ([8], [2], [3]) to extend these inequalities from sums to general functions of independent variables, for which the centered conditional versions are sub-Gaussian or sub-exponential, respectively. These concentration inequalities, Theorem 3, 4 and 5, are stated in Section 3 below. Theorems 4 and 5, which apply to the heavier tailed sub-exponential distributions, are our principal contributions. Theorem 3 for the sub-Gaussian case has less novelty, but it is included to complete the picture, and because its proof provides a good demonstration of the entropy method.

For the purpose of illustration we apply these results to some standard problems in learning theory, vector valued concentration, the generalization of PCA and the method of Rademacher complexities. Over the last twenty years the latter method ([1], [5]) has been successfully used to prove generalization bounds in a variety of situations. The Rademacher complexity itself does not necessitate boundedness, but, when losses and data-distributions are unbounded, the use of the bounded difference inequality can only be circumvented with considerable effort. Using our bounds the extension is immediate. We also show how an inequality of Kontorovich [6], which describes concentration on products of sub-Gaussian metric probability spaces and has applications to algorithmic stability, can be extended to the sub-exponential case.

**Related work** Several works contain results very similar to Theorem 3, which refers to the sub-Gaussian case. The closest to it is Theorem 3 in [12], which gives essentially the same learning bounds for sub-Gaussian distributions. Theorem 1 in [6] is also somewhat similar, but specializes to metric probability spaces. Somewhat akin is the work in [7].

35th Conference on Neural Information Processing Systems (NeurIPS 2021).

To address the sub-exponential case, we have not found results comparable to Theorems 4 and 5 in the literature.

There has been a lot of work to establish generalization in unbounded situations ([12], [4], [6]), or the astounding results in [13], but we are unaware of an equally simple extension of the method of Rademacher complexities to sub-exponential distributions, as the one given below.

## 2 Preliminaries

### 2.1 Notation and conventions

We use upper-case letters for random variables and vectors of random variables and lower case letters for scalars and vectors of scalars. In the sequel $X = (X_1, \ldots, X_n)$ is a vector of independent random variables with values in a space $\mathcal{X}$, the vector $X' = (X'_1, \ldots, X'_n)$ is iid to $X$ and $f$ is a function $f : \mathcal{X}^n \to \mathbb{R}$. We are interested in concentration of the random variable $f(X)$ about its expectation, and require some special notation to describe the fluctuations of $f$ in its $k$-th variable $X_k$, when the other variables $(x_i : i \neq k)$ are given.

**Definition 1** *If $f : \mathcal{X}^n \to \mathbb{R}$, $x = (x_1, ..., x_n) \in \mathcal{X}^n$ and $X = (X_1, ..., X_n)$ is a random vector with independent components in $\mathcal{X}^n$, then the $k$-th centered conditional version of $f$ is the random variable*

$$f_k(X)(x) = f(x_1, \ldots, x_{k-1}, X_k, x_{k+1}, \ldots, x_n) - \mathbb{E}\left[ f(x_1, \ldots, x_{k-1}, X'_k, x_{k+1}, \ldots, x_n) \right].$$

Then $f_k(X)$ is a random-variable-valued function $f_k(X) : x \in \mathcal{X}^n \mapsto f_k(X)(x)$, which does not depend on the $k$-th coordinate of $x$. If $\|.\|_a$ is any given norm on random variables, then $\|f_k(X)\|_a(x) := \|f_k(X)(x)\|_a$ defines a non-negative real-valued function $\|f_k(X)\|_a$ on $\mathcal{X}^n$. Thus $\|f_k(X)\|_a(X)$ is also a random variable, of which $\|\|f_k(X)\|_a\|_\infty$ is the essential supremum. If $X'$ is iid to $X$ then $\|f_k(X)\|_a$ is the same function as $\|f_k(X')\|_a$ and $\|f_k(X)\|_a(X')$ is iid to $\|f_k(X)\|_a(X)$. Note that

$$f_k(X)(X) = f(X) - \mathbb{E}\left[ f(X) | X_1, ..., X_{k-1}, X_{k+1}, ... X_n \right].$$

Also, if $f(x) = \sum_{i=1}^n x_i$, then $f_k(X)(x) = X_k - \mathbb{E}[X_k]$ is independent of $x$.

If $H$ is a Hilbert space, then the Hilbert space of Hilbert-Schmidt operators $HS(H)$ is the set of bounded operators $T$ on $H$ satisfying $\|T\|_{HS} = \sqrt{\sum_{ij} \langle Te_i, e_j \rangle_H^2} < \infty$ with inner product $\langle T, S \rangle_{HS} = \sum_{ij} \langle Te_i, e_j \rangle_H \langle Se_i, e_j \rangle_H$, where $(e_i)$ is an orthonormal basis. For $x \in H$ the operator $Q_x \in HS(H)$ is defined by $Q_x y = \langle y, x \rangle x$, and one verifies that $\|Q_x\|_{HS} = \|x\|_H^2$.

### 2.2 Sub-Gaussian and sub-exponential norms

It follows from Propositions 2.7.1 and 2.5.2 in [14] that we can equivalently redefine the usual sub-Gaussian and sub-exponential norms $\|\cdot\|_{\psi_2}$ and $\|\cdot\|_{\psi_1}$ for any real random variable $Z$ as

$$\|Z\|_{\psi_2} = \sup_{p \geq 1} \frac{\|Z\|_p}{\sqrt{p}} \text{ and } \|Z\|_{\psi_1} = \sup_{p \geq 1} \frac{\|Z\|_p}{p}, \tag{1}$$

where the $L_p$-norms are defined as $\|Z\|_p = \left( \mathbb{E}\left[ |Z|^p \right] \right)^{1/p}$. It also follows from the above mentioned propositions that for every centered sub-Gaussian random variable $Z$ we have, for all $\beta \in \mathbb{R}$,

$$\mathbb{E}\left[ e^{\beta Z} \right] \leq e^{4e\beta^2 \|Z\|_{\psi_2}^2}. \tag{2}$$

Sub-exponential variables ($\|Z\|_{\psi_1} < \infty$) have heavier tails than sub-Gaussian variables and include the exponential, chi-squared and Poisson distributions. Products and squares of sub-Gaussian variables are sub-exponential, in particular

$$\left\| Z^2 \right\|_{\psi_1} = \sup_{p \geq 1} \frac{\left\| Z^2 \right\|_p}{p} = 2 \sup_{p \geq 1} \left( \frac{\|Z\|_{2p}}{\sqrt{2p}} \right)^2 \leq 2 \|Z\|_{\psi_2}^2$$

(we would have $\left\|Z^2\right\|_{\psi_1} = \|Z\|_{\psi_2}^2$, if the norms were defined as in [14]). All sub-Gaussian and bounded variables are sub-exponential. For bounded variables we have $\|Z\|_{\psi_1} \leq \|Z\|_{\psi_2} \leq \|Z\|_\infty$, but for concentrated variables the sub-Gaussian and sub-exponential norms can be much smaller. The arithmetic mean of $N$ iid bounded variables has uniform norm $O(1)$, sub-Gaussian norm $O\left(N^{-1/2}\right)$, and the square of the mean has sub-exponential norm $O\left(N^{-1}\right)$ (see [14]). The inequalities of the next section can therefore be applied successfully to $n$ such variables, even when the bounded difference inequality gives only trivial results, for example when $n < \ln(1/\delta)$, where $\delta$ is the confidence parameter. For strongly concentrated variables we have the following lemma (with proof in the supplement).

**Lemma 2** *Suppose the random variable $X$ satisfies $E[X] = 0$, $|X| \leq 1$ a.s. and $\Pr\{|X| > \epsilon\} \leq \epsilon$ for some $\epsilon > 0$. Then $\forall p \geq 1$, $\|X\|_p \leq 2\epsilon^{1/p}$ and $\|X\|_{\psi_1} \leq 2\left(e \ln(1/\epsilon)\right)^{-1}$.*

In a nearly deterministic situation, with $\epsilon = e^{-d}$, we have $\|X\|_{\psi_1} \leq O(1/d)$, and a simple union bound of the sub-exponential inequalities allows uniform estimation of $e^d$ such variables with sample size $O(n) \leq O(d)$.

In several applications we will require a sub-Gaussian or sub-exponential bound on the norm of a random vector. This may seem quite restrictive. If $X = \sum Z_i$, then in general we can only say $\|\|X\|\|_{\psi_1} \leq \sum_i \|\|Z_i\|\|_{\psi_1}$, so if $\mathcal{X} = \mathbb{R}^d$ with basis $(e_i)$ then our most general estimate is $\|\|X\|\|_{\psi_1} \leq \sum_{i=1}^d \|\langle e_i, X\rangle\|_{\psi_1}$, which has poor dimension dependence. But in many situations in machine learning one can assume that $X$ is a sum, $X = Z_{signal} + Z_{noise}$, where $\|Z_{signal}\|$ is bounded and the perturbing component $\|Z_{noise}\|$ is of small sub-exponential norm, albeit potentially unbounded.

## 3 Results

Our first result assumes sub-Gaussian versions $f_k(X)$. It is an unbounded analogue of the popular bounded difference inequality, which is sometimes also called McDiarmid's inequality ([3], [11]).

**Theorem 3** *Let $f : \mathcal{X}^n \to \mathbb{R}$ and $X = (X_1, \ldots, X_n)$ be a vector of independent random variables with values in a space $\mathcal{X}$. Then for any $t > 0$ we have*

$$\Pr\{f(X) - \mathbb{E}[f(X')] > t\} \leq \exp\left(\frac{-t^2}{32e\left\|\sum_k \|f_k(X)\|_{\psi_2}^2\right\|_\infty}\right).$$

If $f$ is a sum of sub-Gaussian variables this reduces to the general Hoeffding inequality, Theorem 2.6.2 in [14]. On the other hand, if the $f_k(X)$ are a.s. bounded, $\|f_k(X)\|_\infty(x) \leq r_k(x)$, then also $\|f_k(X)\|_{\psi_2}(x) \leq r_k(x)$ and we recover the bounded difference inequality (Theorem 6.5 in [3]) up to a constant factor. A similar result to Theorem 3 is given with better constants in [6], although in specialized and slightly weaker forms, where the essential supremum is inside the sum in the denominator of the exponent.

The next two results are our principal contributions and apply to functions with sub-exponential conditional versions.

**Theorem 4** *With $f$ and $X$ as in Theorem 3 for any $t > 0$*

$$\Pr\{f(X) - \mathbb{E}[f(X')] > t\}$$

$$\leq \exp\left(\frac{-t^2}{4e^2\left\|\sum_k \|f_k(X)\|_{\psi_1}^2\right\|_\infty + 2e\max_k \left\|\|f_k(X)\|_{\psi_1}\right\|_\infty t}\right).$$

The bound exhibits a sub-Gaussian tail governed by the variance-proxy $\left\|\sum_k \|f_k(X)\|_{\psi_1}^2\right\|_\infty$ for small deviations, and a sub-exponential tail governed by the scale-proxy $\max_k \left\|\|f_k(X)\|_{\psi_1}\right\|_\infty$ for large deviations. If $f$ is a sum we recover the inequality in [14], Theorem 2.8.1.

In Theorem 4 both the variance-proxy and the scale proxy depend on the sub-exponential norms $\|\cdot\|_{\psi_1}$. A well known two-tailed bound for sums of bounded variables, Bernstein's inequality [11], has the variance proxy depending on $\|\cdot\|_2$ and the scale-proxy on $\|\cdot\|_\infty$. When $\|\cdot\|_2 \ll \|\cdot\|_\infty$ this leads to tighter bounds, whenever the inequality is operating in the sub-Gaussian regime, which often happens for large sample-sizes. The next result allows a similar use, whenever $\|\cdot\|_{2p} \ll q \|\cdot\|_{\psi_1}$ for conjugate exponents $p$ and $q$.

**Theorem 5** *With $f$ and $X$ as above let $p, q \in (1, \infty)$ satisfy $p^{-1} + q^{-1} = 1$. Then for any $t > 0$*

$$\Pr\{f(X) - \mathbb{E}[f(X')] > t\} \le \exp\left(\frac{-t^2}{2\left\|\sum_k \|f_k(X)\|_{2p}^2\right\|_\infty + 2eq \max_k \left\|\|f_k(X)\|_{\psi_1}\right\|_\infty t}\right).$$

We cannot let $p \to 1$ to recover the behaviour of Bernstein's inequality in the sub-Gaussian regime, because this would drive the scale-proxy to infinity. But already $p = q = 2$ can give substantial improvements over Theorem 4, if the distributions of the $f_k(X)$ are very concentrated. This inequality appears to be new even if applied to sums. A proof is given in the supplement, where we also show, that the $q$ in the scale-proxy can be replaced by $\sqrt{q}$, if the sub-exponential norm is replaced by the sub-Gaussian norm.

We conclude this section with a centering lemma, which will be useful in applications. The proof is given in the supplement.

**Lemma 6** *Let $X, X'$ be iid with values in $\mathcal{X}$, $\phi : \mathcal{X} \times \mathcal{X} \to \mathbb{R}$ measurable, $\alpha \in \{1, 2\}$. Then*

*(i) $\|\mathbb{E}[\phi(X, X')|X]\|_{\psi_\alpha} \le \|\phi(X, X')\|_{\psi_\alpha}$*

*(ii) If $\mathcal{X} = \mathbb{R}$ then $\|X - \mathbb{E}[X]\|_{\psi_\alpha} \le 2\|X\|_{\psi_\alpha}$.*

One consequence of this lemma is, that we could equally well work with uncentered conditional versions, if we adjust the constants by an additional factor of 2.

## 4 Applications

To illustrate the use of these inequalities we give applications to vector valued concentration and different methods to prove generalization bounds. We concentrate mainly on applications of the more novel Theorems 4 and 5. Applications of the the sub-Gaussian inequality can often be substituted by the reader following the same pattern.

### 4.1 Vector valued concentration

We begin with concentration of vectors in a normed space $(\mathcal{X}, \|.\|)$.

**Proposition 7** *Suppose the $X_i$ are independent random variables with values in a normed space $(\mathcal{X}, \|.\|)$ such that $\|\|X_i\|\|_{\psi_1} \le \infty$ and that $\delta > 0$. (i) With probability at least $1 - \delta$*

$$\left\|\sum_i X_i\right\| - \mathbb{E}\left\|\sum_i X_i\right\| \le 4e\sqrt{\sum_k \|\|X_k\|\|_{\psi_1}^2 \ln(1/\delta)} + 4e \max_k \|\|X_k\|\|_{\psi_1} \ln(1/\delta).$$

*The inequality is two-sided, that is the two terms on the left-hand-side may be interchanged.*

*(ii) If $\mathcal{X}$ is a Hilbert space, the $X_i$ are iid, $n \ge \ln(1/\delta) \ge \ln 2$, then with probability at least $1 - \delta$*

$$\left\|\frac{1}{n}\sum_i X_i - \mathbb{E}[X_1']\right\| \le 8e\|\|X_1\|\|_{\psi_1}\sqrt{\frac{2\ln(1/\delta)}{n}}. \tag{3}$$

*(iii) If $\mathcal{X}$ is a Hilbert space, the $X_i$ are iid, $0 < \delta \le 1/2$ and $p, q \in (1, \infty)$ are conjugate exponents then with probability at least $1 - \delta$*

$$\left\|\frac{1}{n}\sum_i X_i - \mathbb{E}[X_1']\right\| \le 2\|\|X_1 - \mathbb{E}[X_1']\|\|_{2p}\sqrt{\frac{2\ln(1/\delta)}{n}} + 4eq\|\|X_1\|\|_{\psi_1}\frac{\ln(1/\delta)}{n}.$$

The purpose of the simple inequality (3) is to give a compact expression, when it is possible to restrict to the sub-Gaussian regime with the assumption $n \geq \ln(1/\delta)$. This is often possible in applications. Part (iii) gives better estimation bounds when the lower order moments $\|\cdot\|_{2p}$ ar small.

**Proof.** (i) We look at the function $f(x) = \|\sum_i x_i\|$. Then

$$|f_k(X)(x)| = \left| \left\| \sum_{i \neq k} x_i + X_k \right\| - \mathbb{E}\left[ \left\| \sum_{i \neq k} x_i + X_k' \right\| \right] \right| \leq \mathbb{E}\left[ \|X_k - X_k'\| \,|\, X \right].$$

Observe that the bound on $f_k(X)(x)$ (not $f_k(X)(x)$ itself) is independent of $x$. Using Lemma 6 we get

$$\|f_k(X)(x)\|_{\psi_1} \leq \|\|X_k - X_k'\|\|_{\psi_1} \leq 2 \|\|X_k\|\|_{\psi_1},$$

and the first conclusion follows from Theorem 4 by equating the probability to $\delta$ and solving for $t$. The proofs of (ii) and (iii) follow a similar pattern and are given in the supplement. ∎

## 4.2 A uniform bound for PCA

With the results of the previous section it is very easy to obtain a uniform bound for principal subspace selection (often called PCA for principal component analysis) with sub-Gaussian data. In PCA we look for a projection onto a $d$-dimensional subspace which most faithfully represents the data. Let $H$ be a Hilbert-space, $X_i$ iid with values in $H$ and $\mathcal{P}_d$ the set of $d$-dimensional orthogonal projection operators in $H$. For $x \in H$ and $P \in \mathcal{P}_d$ the reconstruction error is $\ell(P,x) := \|Px - x\|_H^2$. We give a bound on the estimation difference between the expected and the empirical reconstruction error, uniform for projections in $\mathcal{P}_d$.

**Theorem 8** *With $X = (X_1, ..., X_n)$ iid and $n \geq \ln(1/\delta) \geq \ln 2$ we have with probability at least $1 - \delta$*

$$\sup_{P \in \mathcal{P}_d} \frac{1}{n} \sum_i \mathbb{E}[\ell(P, X_1)] - \ell(P, X_i) \leq 16e\left(\sqrt{d} + 1\right) \|\|X_1\|\|_{\psi_2}^2 \sqrt{\frac{2\ln(2/\delta)}{n}}.$$

**Proof.** It is convenient to work in the space of Hilbert-Schmidt operators $HS(H)$, where we can write $\ell(P,x) = \|Q_x\|_{HS} - \langle P, Q_x \rangle_{HS}$. Then

$$\sup_{P \in \mathcal{P}_d} \frac{1}{n} \sum_i \mathbb{E}[\ell(P, X_1)] - \ell(P, X_i)$$

$$= \sup_{P \in \mathcal{P}_d} \left\langle P, \frac{1}{n} \sum_i (Q_{X_i} - \mathbb{E}[Q_{X_i}]) \right\rangle_{HS} + \left( \mathbb{E}\|Q_{X_i}\|_{HS} - \frac{1}{n} \sum_i \|Q_{X_i}\|_{HS} \right).$$

Since for $P \in \mathcal{P}_d$ we have $\|P\|_{HS} = \sqrt{d}$, we can use Cauchy-Schwarz and Proposition 7 (ii) to bound the first term above with probability at least $1 - \delta$ as

$$\sqrt{d} \left\| \frac{1}{n} \sum_i (Q_{X_i} - \mathbb{E}[Q_{X_1}]) \right\|_{HS} \leq 8e\sqrt{d} \|\|Q_{X_1}\|_{HS}\|_{\psi_1} \sqrt{\frac{2\ln(1/\delta)}{n}}.$$

The remaining term is bounded by applying the same result to the random vectors $\|Q_{X_i}\|_{HS}$ in the Hilbert space $\mathbb{R}$ (note that this just involves a sum of sub-exponential variables and could already be handled with Theorem 2.8.1 in [14]). The result follows from combining both bounds in a union bound and noting that $\|\|Q_{X_1}\|_{HS}\|_{\psi_1} = \|\|X_1\|^2\|_{\psi_1} \leq 2 \|\|X_1\|\|_{\psi_2}^2$. ∎

## 4.3 Generalization with Rademacher complexities

Suppose that $\mathcal{H}$ is a class of functions $h : \mathcal{X} \to \mathbb{R}$. We seek a high-probability bound on the supremum deviation, which is the random variable

$$f(X) = \sup_{h \in \mathcal{H}} \frac{1}{n} \sum_{i=1}^n (h(X_i) - \mathbb{E}[h(X_i')]).$$

The now classical method of Rademacher complexities ([1], [5]) writes $f(X)$ as the sum

$$f(X) = (f(X) - \mathbb{E}[f(X')]) + \mathbb{E}[f(X')] \tag{4}$$

and bounds the two terms separately. The first term is bounded using a concentration inequality, the second term $\mathbb{E}[f(X)]$ is bounded by symmetrization. If the $\epsilon_i$ are independent Rademacher variables, uniformly distributed on $\{-1, 1\}$, then

$$\mathbb{E}[f(X)] \leq \mathbb{E}\left[\frac{2}{n}\mathbb{E}\left[\sup_{h \in \mathcal{H}} \sum_i \epsilon_i h(X_i) \,|X\right]\right] =: \mathbb{E}[\mathcal{R}(\mathcal{H}, X)].$$

Further bounds on this quantity depend on the class in question, but they do not necessarily require the $h(X_i)$ to be bounded random variables, Lipschitz properties being more relevant. For the first term $f(X) - \mathbb{E}[f(X)]$, however, the classical approach uses the bounded difference inequality, which requires boundedness. We now show that boundedness can be replaced by sub-exponential distributions for uniformly Lipschitz function classes.

**Theorem 9** *Let $X = (X_1, ..., X_n)$ be iid random variables with values in a Banach space $(\mathcal{X}, \|\cdot\|)$ and let $\mathcal{H}$ be a class of functions $h : \mathcal{X} \to \mathbb{R}$ such that $h(x) - h(y) \leq L\|x - y\|$ for all $h \in \mathcal{H}$ and $x, y \in \mathcal{X}$. If $n \geq \ln(1/\delta)$ then with probability at least $1 - \delta$*

$$\sup_{h \in \mathcal{H}} \frac{1}{n} \sum_i h(X_i) - \mathbb{E}(h(X)) \leq \mathbb{E}[\mathcal{R}(\mathcal{H}, X)] + 16eL \,\|\|X_1\|\|_{\psi_1} \sqrt{\frac{\ln(1/\delta)}{n}}.$$

**Proof.** The vector space

$$\mathcal{B} = \left\{g : \mathcal{H} \to \mathbb{R} : \sup_{h \in \mathcal{H}} |g(h)| < \infty\right\}$$

becomes a normed space with norm $\|g\|_{\mathcal{B}} = \sup_{h \in \mathcal{H}} |g(h)|$. For each $X_i$ define $\hat{X}_i \in \mathcal{B}$ by $\hat{X}_i(h) = (1/n)(h(X_i) - \mathbb{E}[h(X'_i)])$. Then the $\hat{X}_i$ are zero mean random variables in $\mathcal{B}$ and $f(X) = \left\|\sum_i \hat{X}_i\right\|_{\mathcal{B}}$. Also with Lemma 6 and the iid-assumption

$$\begin{aligned}
\left\|\left\|\hat{X}_i\right\|_{\mathcal{B}}\right\|_{\psi_\alpha} &= \frac{1}{n}\left\|\sup_h (\mathbb{E}[h(X_i) - h(X'_i)]\,|X)\right\|_{\psi_\alpha} \\
&\leq \frac{L}{n}\|\mathbb{E}[\|X_i - X'_i\|]\,|X\|_{\psi_\alpha} \leq \frac{2L}{n}\,\|\|X_1\|\|_{\psi_\alpha},
\end{aligned}$$

and from Proposition 7 (ii) we get with probability at least $1 - \delta$

$$f(X) - \mathbb{E}[f(X')] \leq 16eL\,\|\|X_1\|\|_{\psi_1}\sqrt{\frac{\ln(1/\delta)}{n}}.$$

The result follows from (4). ∎

**Remarks.** 1. A possible candidate for $\mathcal{H}$ would be a ball of radius $L$ in the dual space $\mathcal{X}^*$, composed with Lipschitz functions, like the hinge-loss.

2. If, instead of using Proposition 7, one directly considers the centered conditional versions of $f(X)$, the constants above can be improved at the expense of a slightly more complicated proof.

3. A corresponding sub-Gaussian result can be supplied along the same lines by using Theorem 3 instead of Theorem 4. Such a result has been given in [12], Theorem 3, using a sub-Gaussian condition which involves the supremum over the function class. The sub-exponential bound above is new as far as we know, and in the relevant regime $n > \ln(1/\delta)$ it improves over the sub-Gaussian case, since $\|\|X_1\|\|_{\psi_1} \leq \|\|X_1\|\|_{\psi_2}$.

As a concrete case consider linear regression with potentially unbounded data. Let $\mathcal{X} = (H, \mathbb{R})$, where $H$ is a Hilbert-space with inner product $\langle ., . \rangle$ and norm $\|.\|_H$, and let $X_1$ and $Z_1$ be each sub-exponential random variables in $H$ and $\mathbb{R}$ respectively. The pair $(X_1, Z_1)$ represents the joint occurrence of input-vectors $X_1$ and real outputs $Z_1$. On $\mathcal{X}$ we consider the class $\mathcal{H}$ of functions $\mathcal{H} = \{(x, z) \mapsto h(x, z) = \ell(\langle w, x \rangle - z) : \|w\|_H \leq L\}$, where $\ell$ is a 1-Lipschitz loss function, like the absolute error or the Huber loss.

**Corollary 10** *Let $\mathcal{X}$ and $\mathcal{H}$ be as above and $(X, Z) = ((X_1, Z_1), ..., (X_n, Z_n))$ be an iid sample of random variables in $\mathcal{X}$. Then for $\delta > 0$ and $n \geq \ln(1/\delta)$ with probability at least $1 - \delta$*

$$\sup_{h \in \mathcal{H}} \frac{1}{n} \sum_i h(X_i, Z_i) - \mathbb{E}(h(X_i, Z_i)) \leq \frac{8}{\sqrt{n}} \left( L \|\|X_1\|\|_{\psi_1} + \|\|Z_1\|\|_{\psi_1} \right) \left( 1 + 2e\sqrt{\ln(1/\delta)} \right).$$

The proof of this corollary is given in the supplement.

### 4.4 Unbounded metric spaces and algorithmic stability

We use Theorem 2 to extend a method of Kontorovich [6] from sub-Gaussian to sub-exponential distributions. If $(\mathcal{X}, d, \mu)$ is a metric probability space and $X, X' \sim \mu$ are iid random variables with values in $\mathcal{X}$, Kontorovich defines the sub-Gaussian diameter of $(\mathcal{X}, d)$ as the optimal sub-Gaussian parameter of the random variable $\epsilon d(X, X')$, where $\epsilon$ is a Rademacher variable. The Rademacher variable is needed in [6] to work with centered random variables, which gives better constants. In our case we work with norms and we can more simply define the sub-Gaussian and sub-exponential diameters respectively as

$$\Delta_\alpha(\mathcal{X}, d, \mu) = \|d(X, X')\|_{\psi_\alpha} \text{ for } \alpha \in \{1, 2\} \text{ and independent } X', X \sim \mu.$$

Then Theorem 4 implies the following result, the easy proof of which is given in the supplement.

**Theorem 11** *For $1 \leq i \leq n$ let $X_i$ be independent random variables distributed as $\mu_i$ in $\mathcal{X}$, $X = (X_1, ..., X_n)$, $X'$ iid to $X$, and let $f : \mathcal{X}^n \to \mathbb{R}$ have Lipschitz constant $L$ with respect to the metric $\rho$ on $\mathcal{X}^n$ defined by $\rho(x, y) = \sum_i d(x_i, y_i)$. Then for $t > 0$*

$$\Pr\{f(X) - \mathbb{E}[f(X')] > t\} \leq \exp\left( \frac{-t^2}{4eL^2 \sum_i \Delta_1(\mathcal{X}, d, \mu_i)^2 + 2e \max_i \Delta_1(\mathcal{X}, d, \mu_i) t} \right).$$

This is the sub-exponential counterpart to Theorem 1 of [6], a version of which could have been derived using Theorem 3 in place of 4. Our result can be equally substituted to establish generalization using the notion of total Lipschitz stability, just as in [6]. We also note that Theorem 4 of the latter work also gives bounds for different Orlicz norms $\|.\|_{\psi_p}$, but it requires $p > 1$, and the bounds deteriorate as $p \to 1$.

## 5 Proofs of Theorems 3 and 4

We first collect some necessary tools. Central to the entropy method is the entropy $S(Y)$ of a real valued random variable $Y$ defined as

$$S(Y) = \mathbb{E}_Y[Y] - \ln \mathbb{E}\left[e^Y\right],$$

where the tilted expectation $\mathbb{E}_Y$ is defined as $\mathbb{E}_Y[Z] = \mathbb{E}\left[Ze^Y\right] / \mathbb{E}\left[e^Y\right]$. Using Theorem 1 in [10] the logarithm of the moment generating function can be expressed in terms of the entropy as

$$\ln \mathbb{E}\left[e^{\beta(Y - \mathbb{E}[Y])}\right] = \beta \int_0^\beta \frac{S(\gamma Y) \, d\gamma}{\gamma^2}. \tag{5}$$

If $f : \mathcal{X}^n \to \mathbb{R}$ and $X$ and the $f_k$ are as in the introduction then the conditional entropy is the function $S_{f,k} : \mathcal{X}^n \to \mathbb{R}$ defined by $S_{f,k}(x) = S(f_k(X)(x))$ for $x \in \mathcal{X}^n$. At the heart of the method is the sub-additivity of entropy (Theorem 6 in [10] or Theorem 4.22 in [3])

$$S(f(X)) \leq \mathbb{E}_{f(X)}\left[\sum_{i=1}^n S_{f,k}(X)\right]. \tag{6}$$

The following lemma gives a bound on the entropy of a sub-Gaussian random variable.

**Lemma 12** *For any centered random variable $Y$ we have (i) $S(Y) \leq \ln \mathbb{E}\left[e^{2Y}\right]$. (ii) If $Y$ is sub-Gaussian and $\beta$ is real then $S(\beta Y) \leq 16e\beta^2 \|Y\|_{\phi_2}^2$.*

**Proof.**

$$S\left(Y\right) = \mathbb{E}_Y\left[\ln\left(\frac{e^Y}{\mathbb{E}\left[e^Y\right]}\right)\right] \le \ln\mathbb{E}_Y\left[\frac{e^Y}{\mathbb{E}\left[e^Y\right]}\right] = \ln\mathbb{E}\left[e^{2Y}\right] - 2\ln\mathbb{E}\left[e^Y\right]$$

$$\le \ln\mathbb{E}\left[e^{2Y}\right].$$

The first inequality follows from Jensen's inequality by concavity of the logarithm, the second by convexity of the exponential function. This gives (i). For (ii) replace $Y$ by $\beta Y$ and use (2) to get $S\left(\beta Y\right) \le \ln\mathbb{E}\left[e^{2\beta Y}\right] \le 16e\beta^2\left\|Y\right\|_{\psi_2}^2$. ∎

**Proof of Theorem 3.** For any $x \in \mathcal{X}^n$ and $\gamma \in \mathbb{R}$ part (ii) of the previous lemma gives $S_{\gamma f,k}\left(x\right) = S\left(\gamma f_k\left(X\right)\left(x\right)\right) \le 16e\gamma^2\left\|f_k\left(X\right)\left(x\right)\right\|_{\psi_2}^2$. By subadditivity of entropy (6) this gives

$$S\left(\gamma f\left(X\right)\right) \le 16e\gamma^2\mathbb{E}_{\gamma f(X)}\left[\sum_k\left\|f_k\left(X'\right)\right\|_{\psi_2}^2\left(X\right)\right] \le 16e\gamma^2\left\|\sum_k\left\|f_k\left(X\right)\right\|_{\psi_2}^2\right\|_\infty.$$

Using Markov's inequality and (5) this gives

$$\Pr\left\{f\left(X\right) - \mathbb{E}\left[f\left(X'\right)\right] > t\right\} \le \exp\left(\beta\int_0^\beta\frac{S\left(\gamma f\left(X\right)\right)d\gamma}{\gamma^2} - \beta t\right)$$

$$\le \exp\left(16e\beta^2\left\|\sum_k\left\|f_k\left(X\right)\right\|_{\psi_2}^2\right\|_\infty d\gamma - \beta t\right).$$

Minimization in $\beta$ concludes the proof. ∎

Lemma 12 (i) and the preceeding proof provide a general template to convert many exponential tail-bounds for sums into analogous bounds for general functions. For sums $\sum X_i$ one typically has a bound on $\ln\mathbb{E}\left[e^{\beta X_i}\right]$. Lemma 12 then provides an analogous bound on the entropy of the conditional versions of a general function, and subadditivity of entropy and (5) complete the conversion, albeit with a deterioration of constants. Using part (v) of Proposition 2.7.1 in [14] this method would lead to an easy proof of Theorem 4, in a form exactly like Theorem 2.8.1 [14]. Here we will use a slightly different method which gives better constants and will also provide the proof of Theorem 5.

For the proof of Theorem 4 we use the following fluctuation representation of entropy (Theorem 3 in [10]).

$$S\left(Y\right) = \int_0^1\left(\int_t^1\mathbb{E}_{sY}\left[\left(Y - \mathbb{E}_{sY}\left[Y\right]\right)^2\right]ds\right)dt \tag{7}$$

We use this to bound the entropy of a centered sub-exponential random variable.

**Lemma 13** *If* $\left\|Y\right\|_{\psi_1} < 1/e$ *and* $\mathbb{E}\left[Y\right] = 0$ *then*

$$S\left(Y\right) \le \frac{e^2\left\|Y\right\|_{\psi_1}^2}{\left(1 - e\left\|Y\right\|_{\psi_1}\right)^2}.$$

**Proof.** Let $s \in [0,1]$

$$\mathbb{E}_{sY}\left[\left(Y - \mathbb{E}_{sY}\left[Y\right]\right)^2\right] \le \mathbb{E}_{sY}\left[Y^2\right] = \frac{\mathbb{E}\left[Y^2e^{sY}\right]}{\mathbb{E}\left[e^{sY}\right]} \le \mathbb{E}\left[Y^2e^{sY}\right].$$

The first inequality follows from the variational property of variance, the second from Jensen's inequality since $\mathbb{E}\left[Y_k\right] = 0$. Expanding the exponential we get

$$\mathbb{E}\left[Y^2e^{sY}\right] \le \mathbb{E}\left[\sum_{m=0}^\infty\frac{s^m}{m!}Y^{m+2}\right] = \sum_{m=0}^\infty\frac{s^m}{m!}\mathbb{E}\left[Y^{m+2}\right].$$

The interchange of expectation and summation will be justified by absolute convergence of the sum as follows.

$$
\begin{aligned}
\sum_{m=0}^{\infty} \frac{s^m}{m!} \mathbb{E}\left[Y^{m+2}\right] &\leq \sum_{m=0}^{\infty} \frac{s^m}{m!} \|Y\|_{\psi_1}^{m+2} (m+2)^{m+2} \\
&\leq e^2 \|Y\|_{\psi_1}^2 \sum_{m=0}^{\infty} (m+2)(m+1) \left(se\|Y\|_{\psi_1}\right)^m.
\end{aligned}
$$

The first inequality follows from definition of $\|Y\|_{\psi_1}$, and the second from Stirling's approximation $(m+2)^{m+2} \leq (m+2)! e^{m+2}$. Absolute convergence is insured since $se\|Y\|_{\psi_1} < 1$. Using

$$
\int_0^1 \int_t^1 s^m ds\, dt = \frac{1}{m+2}
$$

the fluctuation representation (7) and the above inequalities give

$$
\begin{aligned}
S(Y) &= \int_0^1 \left(\int_t^1 \mathbb{E}_{sY}\left[(Y - \mathbb{E}_{sY}[Y])^2\right] ds\right) dt \\
&\leq e^2 \|Y\|_{\psi_1}^2 \sum_{m=0}^{\infty} (m+1)\left(e\|Y\|_{\phi_1}\right)^m = \frac{e^2 \|Y\|_{\psi_1}^2}{\left(1 - e\|Y\|_{\psi_1}\right)^2}.
\end{aligned}
$$

∎

We also need the following lemma (Lemma 12 in [9]).

**Lemma 14** *Let $C$ and $b$ denote two positive real numbers, $t > 0$. Then*

$$
\inf_{\beta \in [0, 1/b)} \left(-\beta t + \frac{C\beta^2}{1 - b\beta}\right) \leq \frac{-t^2}{2(2C + bt)}. \tag{8}
$$

**Proof of Theorem 4.** We abbreviate $M := \max_k \left\|\|f_k(X)\|_{\psi_1}\right\|_{\infty}$ and let $0 < \gamma \leq \beta < (eM)^{-1}$. Then for any $x \in \mathcal{X}^n$ and $k \in \{1, ..., n\}$ we have $\|\gamma f_k(X)(x)\|_{\psi_1} < \|f_k(X)(x)\|_{\psi_1} / (eM) \leq 1/e$ by the definition of $M$. We can therefore apply the previous lemma to $\gamma f_k(X)(x)$. It gives for almost all $x$

$$
S_{\gamma f, k}(x) = S(\gamma f_k(X)(x)) \leq \frac{e^2 \|\gamma f_k(X)(x)\|_{\psi_1}^2}{\left(1 - e\|\gamma f_k(X)(x)\|_{\psi_1}\right)^2} \leq \frac{\gamma^2 e^2 \|f_k(X)(x)\|_{\psi_1}^2}{(1 - \gamma eM)^2}
$$

Subadditivity of entropy (6) then yields the total entropy bound

$$
\begin{aligned}
S(\gamma f(X)) &\leq \mathbb{E}_{\gamma f(X)}\left[\sum_k S_{\gamma f, k}(X)\right] \leq \frac{\gamma^2 e^2 \mathbb{E}_{\gamma f(X)}\left[\sum_k \|f_k(X')\|_{\psi_1}^2 (X)\right]}{(1 - \gamma eM)^2} \\
&\leq \frac{\gamma^2 e^2 \left\|\sum_k \|f_k(X)\|_{\psi_1}^2\right\|_{\infty}}{(1 - \gamma eM)^2}.
\end{aligned} \tag{9}
$$

Together with (5) this gives

$$
\ln \mathbb{E}\left[e^{\beta(f - \mathbb{E}f)}\right] = \beta \int_0^\beta \frac{S(\gamma f(X))\, d\gamma}{\gamma^2} \leq \frac{\beta^2 e \left\|\sum_k \|f_k(X)\|_{\psi_1}^2\right\|_{\infty}}{1 - \beta eM},
$$

and the concentration inequality then follows from Markov's inequality and Lemma 14. ∎

## 6 Conclusion

In this paper, we presented an extension of Hoeffding- and Bernstein-type inequalities for sums of sub-Gaussian and sub-exponential independent random variables to general functions, and illustrated these inequalities with applications to statistical learning theory.

We hope that future work will reveal other interesting applications of these inequalities.

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
