# A Appendix

## A.1 Remaining proofs for Section 3

We give the missing proof for Theorem 5. The next lemma replaces Lemma 13.

**Lemma 15** *Let $\mathbb{E}[Y] = 0$ and $1 < p, q < \infty$ be conjugate exponents $(1/p + 1/q = 1)$. If $\|Y\|_{\psi_1} < 1/(eq)$ then*

$$S(Y) \leq \frac{\|Y^2\|_p}{2\left(1 - eq\|Y\|_{\psi_1}\right)^2}.$$

*If $\|Y\|_{\psi_2} < 1/\left(e\sqrt{q}\right)$ the same inequality holds with $q\|Y\|_{\psi_1}$ replaced by $\sqrt{q}\|Y\|_{\psi_2}$.*

**Proof.** As in the proof of Lemma 13 we let $s \in [0, 1]$ and obtain the inequality

$$\mathbb{E}_{sY}\left[(Y - \mathbb{E}_{sY}[Y])^2\right] \leq \mathbb{E}\left[\sum_{m=0}^{\infty} \frac{s^m}{m!} Y^{m+2}\right] \leq \|Y^2\|_p \sum_{m=0}^{\infty} \frac{s^m}{m!} \|Y^m\|_q,$$

where the second bound follows from Höler's inequality. Using the definition of $\|.\|_{\psi_1}$ and Stirling's approximation give the bound

$$\|Y^m\|_q = \|Y\|_{mq}^m \leq \left(qm\|Y\|_{\psi_1}\right)^m = m^m\left(q\|Y\|_{\psi_1}\right)^m \leq m!\left(eq\|Y\|_{\psi_1}\right)^m,$$

whence, since $\|Y\|_{\psi_1} < 1/(eq) \implies eq\|Y\|_{\psi_1} < 1$,

$$\sum_{m=0}^{\infty} \frac{s^m}{m!} \|Y^m\|_q \leq \sum_{m=0}^{\infty} \left(s\,eq\|Y\|_{\psi_1}\right)^m \leq \sum_{m=0}^{\infty} \left(eq\|Y\|_{\psi_1}\right)^m \leq \frac{1}{1 - eq\|Y\|_{\psi_1}} < \frac{1}{\left(1 - eq\|Y\|_{\psi_1}\right)^2}.$$

Substitution above gives

$$\mathbb{E}_{sY}\left[(Y - \mathbb{E}_{sY}[Y])^2\right] \leq \|Y^2\|_p \sum_{m=0}^{\infty} \frac{s^m}{m!} \|Y^m\|_q < \frac{\|Y^2\|_p}{\left(1 - eq\|Y\|_{\psi_1}\right)^2},$$

and the double integral in (7) then provides the factor of $1/2$. For the remaining statement repeat the proof and use

$$\|Y\|_{mq}^m \leq \left(\sqrt{qm}\|Y\|_{\psi_2}\right)^m \leq m^m\left(\sqrt{q}\|Y\|_{\psi_2}\right)^m.$$

∎

**Proof of Theorem 5.** We abbreviate $M := \max_k \left\|\|f_k(X)\|_{\psi_1}\right\|_{\infty}$ and let $0 < \gamma \leq \beta < (eqM)^{-1}$. Then for any $k \in \{1, ..., n\}$ we have $\|\gamma f_k(X)\|_{\psi_1} < \|f_k(X)\|_{\psi_1}/(eqM) \leq 1/(eq)$ by the definition of $M$. We can therefore apply the previous Lemma to the random variable $\gamma f_k(X)$. It gives almost surely

$$S(\gamma f_k(X)) \leq \frac{\left\|(\gamma f_k(X))^2\right\|_p}{2\left(1 - eq\|\gamma f_k(X)\|_{\psi_1}\right)^2} \leq \frac{\gamma^2\left\|f_k(X)^2\right\|_p}{2(1 - \gamma eqM)^2}$$

Subadditivity of entropy (6) then yields the total entropy bound

$$
\begin{aligned}
S(\gamma f(X)) &\leq \mathbb{E}_{\gamma f}\left[\sum_k S(\gamma f_k(X))(X)\right] \leq \frac{\gamma^2 \mathbb{E}_{\gamma f(X)}\left[\sum_k \left\|f_k(X)^2\right\|_p(X)\right]}{2(1 - \gamma eqM)^2} \\
&\leq \frac{\gamma^2 \left\|\sum_k \left\|f_k(X)^2\right\|_p\right\|_{\infty}}{2(1 - \gamma eqM)^2}.
\end{aligned}
$$

Together with (5) this gives

$$\ln \mathbb{E}\left[e^{\beta(f-\mathbb{E}f)}\right] = \beta \int_0^\beta \frac{S\left(\gamma f\left(X\right)\right)d\gamma}{\gamma^2} \leq \frac{\beta^2 \left\|\left\|\sum_k \left\|f_k\left(X\right)^2\right\|\right\|_p\right\|_\infty}{2\left(1-\beta eqM\right)},$$

and the concentration inequality then follows from Markov's inequality and Lemma 14, if we set $C = \left\|\left\|\sum_k \left\|f_k\left(X\right)^2\right\|\right\|_p\right\|_\infty /2$ and $b = eqM$. ∎

**Proof of Lemma 6.** By Jensen's inequality for $p \geq 1$

$$
\begin{aligned}
\mathbb{E}\left[\left|\mathbb{E}\left[\phi\left(X,X'\right)|X\right]\right|^p\right] &\leq \mathbb{E}\left[\mathbb{E}\left[\left|\phi\left(X,X'\right)\right||X\right]^p\right] = \mathbb{E}\left[\mathbb{E}\left[\left(\left|\phi\left(X,X'\right)\right|^p\right)^{1/p}|X\right]^p\right] \\
&\leq \mathbb{E}\left[\mathbb{E}\left[\left|\phi\left(X,X'\right)\right|^p|X\right]\right] = \mathbb{E}\left[\left|\phi\left(X,X'\right)\right|^p\right].
\end{aligned}
$$

Therefore $\left\|\mathbb{E}\left[\phi\left(X,X'\right)|X\right]\right\|_p \leq \left\|\phi\left(X,X'\right)\right\|_p$ and (i) follows from our definition of the two norms. If $\mathcal{X} = \mathbb{R}$ and $\phi\left(s,t\right) = s - t$ we get from (i) that

$$\left\|X - \mathbb{E}\left[X'\right]\right\|_{\psi_\alpha} = \left\|\mathbb{E}\left[X - X'|X\right]\right\|_{\psi_\alpha} \leq \left\|X - X'\right\|_{\psi_\alpha} \leq 2\left\|X\right\|_{\psi_\alpha}.$$

∎

## A.2 Remaining proofs for Section 4

Here is a proof of Lemma 2

**Proof of Lemma 2.** For $p \geq 1$ the function $g\left(t\right) = t^{1/p}$ is concave and $g'\left(t\right) = \frac{1}{p}t^{\frac{1}{p}-1}$. It follows that for $s, t \geq 0$

$$\left(t+s\right)^{1/p} \leq t^{1/p} + \frac{st^{\frac{1}{p}-1}}{p} = t^{1/p}\left(1 + \frac{s}{pt}\right) \leq t^{1/p}\left(1 + \frac{s}{t}\right).$$

Under the conditions on $X$ we therefore have

$$
\begin{aligned}
\left\|X\right\|_p &\leq \left(\epsilon + \epsilon^p\left(1-\epsilon\right)\right)^{1/p} \leq \epsilon^{1/p}\left(1 + \frac{\epsilon^p\left(1-\epsilon\right)}{\epsilon}\right) \\
&\leq \epsilon^{1/p}\left(1 + \epsilon^{p-1}\right) \leq 2\epsilon^{1/p}.
\end{aligned}
$$

This proves the first claim. Calculus shows, that the function $p \mapsto 2\epsilon^{1/p}/p$ attains its maximum at $p = \ln\left(1/\epsilon\right)$, so

$$\left\|X\right\|_{\psi_1} = \sup_{p \geq 1} \frac{\left\|X\right\|_p}{p} \leq \sup_{p \geq 1} \frac{2\epsilon^{1/p}}{p} = \frac{2}{e\ln\left(1/\epsilon\right)}.$$

∎

We prove parts (ii) and (iii) of Proposition 7.

**Proof.** (ii) If $\mathcal{X}$ is a Hilbert space and the $X_i$ are iid, then by Jensen's inequality

$$\mathbb{E}\left[\left\|\sum X_i - \mathbb{E}\left[X_i'\right]\right\|\right] \leq \sqrt{n\mathbb{E}\left[\left\|X_1 - \mathbb{E}\left[X_i'\right]\right\|^2\right]} = \sqrt{n}\left\|\left\|X_1\right\|\right\|_2 \leq 2\sqrt{n}\left\|\left\|X_1\right\|\right\|_{\psi_1}. \quad (10)$$

Now let $f\left(x\right) = \left\|\sum_i \left(x_i - \mathbb{E}\left[X_1'\right]\right)\right\|$. Then as in the proof of (i)

$$\left|f_k\left(X\right)\left(x\right)\right| = \left|\left\|\sum_{i \neq k} x_i + X_k - n\mathbb{E}\left[X_1'\right]\right\| - \mathbb{E}\left[\left\|\sum_{i \neq k} x_i + X_k' - n\mathbb{E}\left[X_1'\right]\right\|\right]\right| \leq \mathbb{E}\left[\left\|X_k - X_k'\right\||X\right].$$

and Lemma 6 and Theorem 4 give with probability at least $1 - \delta$

$$
\left\| \sum_i X_i - \mathbb{E}\left[X_1'\right] \right\| \leq \mathbb{E}\left[ \left\| \sum X_i - \mathbb{E}\left[X_i'\right] \right\| \right] + 4e \left\|\|X_1\|\right\|_{\psi_1} \sqrt{n \ln(1/\delta)} + 4e \left\|\|X_1\|\right\|_{\psi_1} \ln(1/\delta)
$$

$$
\leq 2\sqrt{n} \left\|\|X_1\|\right\|_{\psi_1} + 4e \left\|\|X_1\|\right\|_{\psi_1} \sqrt{n \ln(1/\delta)} + 4e \left\|\|X_1\|\right\|_{\psi_1} \ln(1/\delta)
$$

$$
\leq \sqrt{n} \left\|\|X_1\|\right\|_{\psi_1} \left( 2 + 8e\sqrt{\ln(1/\delta)} \right)
$$

$$
\leq 8e\sqrt{n} \left\|\|X_1\|\right\|_{\psi_1} \sqrt{2 \ln(1/\delta)},
$$

where the second inequality follows from (10), the third from $n \geq \ln(1/\delta)$, and the last from $\ln(1/\delta) > \ln 2$. Division by $n$ completes the proof of (ii).

(iii) Apply Theorem 5 to $f(x) = \|\sum_i (x_i - \mathbb{E}\left[X_1'\right])\|$ and solve for the deviation to arrive at

$$
\left\| \sum_i X_i - \mathbb{E}\left[X_1'\right] \right\| \leq \mathbb{E}\left[ \left\| \sum X_i - \mathbb{E}\left[X_i'\right] \right\| \right] + \left\|\|X_1 - \mathbb{E}\left[X_1'\right]\|\right\|_{2p} \sqrt{2n \ln(1/\delta)} + 2eq \left\|\|X_1 - \mathbb{E}\left[X_1'\right]\|\right\|_{\psi_1} \ln(1/\delta)
$$

$$
\leq \sqrt{n} \left\|\|X_1 - \mathbb{E}\left[X_1'\right]\|\right\|_{2p} \left( 1 + \sqrt{2 \ln(1/\delta)} \right) + 4eq \left\|\|X_1\|\right\|_{\psi_1} \ln(1/\delta),
$$

where in the second inequality we bounded the last term using Lemma 6 and the first term with Jensen's inequality as

$$
\mathbb{E}\left[ \left\| \sum X_i - \mathbb{E}\left[X_i'\right] \right\| \right] \leq \sqrt{n} \left\|\|X_1 - \mathbb{E}\left[X_i'\right]\|\right\|_2 \leq \left\|\|X_1 - \mathbb{E}\left[X_1'\right]\|\right\|_{2p},
$$

since $p > 1$. The result follows from using $\delta \leq 1/2$ and division by $n$. ∎

We now prove the Corollary applying to linear regression.

**Proof of Corollary 10.**

$\mathcal{X}$ becomes a Banach space with the norm $\|(x,z)\| = L \|x\|_H + |z|$. Evidently $\left\|\|(X_1, Z_1)\|\right\|_{\psi_1} \leq L \left\|\|X_1\|\right\|_{\psi_1} + \left\|\|Z_1\|\right\|_{\psi_1}$. Then for $h \in \mathcal{H}$

$$
h(x,z) - h(x',z') = \ell(\langle w, x \rangle - z) - \ell(\langle w, x' \rangle - z')
$$
$$
\leq L \|x - x'\|_H + |z - z'| \leq \|(x,z) - (x',z')\|,
$$

so $\mathcal{H}$ is uniformly Lipschitz with constant 1. Also for an iid sample $(X, Z) \in \mathcal{X}^n$ using the Lipschitz property of $\ell$, the triangle inequality and Jensen's inequality, it is not hard to see that

$$
\mathcal{R}(\mathcal{H}, (X, Z)) \leq \frac{2}{n} \left( L \sqrt{\sum_i \|X_i\|_H^2} + \sqrt{\sum_i |Z_i|^2} \right).
$$

Using the iid assumption and $\|\cdot\|_2 \leq 2 \|\cdot\|_{\psi_1}$ we get

$$
\mathbb{E}\left[ \mathcal{R}(\mathcal{H}, (X, Z)) \right] \leq \frac{8}{\sqrt{n}} \left( L \left\|\|X_1\|\right\|_{\psi_1} + \left\|\|Z_1\|\right\|_{\psi_1} \right).
$$

Substitution in Theorem 9 gives for $n \geq \ln(1/\delta)$ with probability at least $1 - \delta$

$$
\sup_{h \in \mathcal{H}} \frac{1}{n} \sum_i h(X_i, Z_i) - \mathbb{E}(h(X_i, Z_i)) \leq \frac{8}{\sqrt{n}} \left( L \left\|\|X_1\|\right\|_{\psi_1} + \left\|\|Z_1\|\right\|_{\psi_1} \right) \left( 1 + 2e\sqrt{\ln(1/\delta)} \right).
$$
∎

Finally we prove the Theorem referring to metric probability spaces.

**Proof of Theorem 11.** The result follows easily from Theorem 4 and

$$
\|f_k(X)(x)\|_{\psi_1}
$$
$$
= \|f(x_1, \ldots, x_{k-1}, X_k, x_{k+1}, \ldots, x_n) - \mathbb{E}[f(x_1, \ldots, x_{k-1}, X_k, x_{k+1}, \ldots, x_n)]\|_{\psi_1}
$$
$$
= \|\mathbb{E}[f(x_1, \ldots, x_{k-1}, X_k, x_{k+1}, \ldots, x_n) - f(x_1, \ldots, x_{k-1}, X_k', x_{k+1}, \ldots, x_n) | X_k]\|_{\psi_1}
$$
$$
\leq L \|\mathbb{E}[d(X_k, X_k') | X_k]\|_{\psi_1}
$$
$$
\leq L \|d(X, X')\|_{\psi_1} = L\Delta_1(\mathcal{X}, d),
$$

where Lemma 6 is used in the last inequality. ∎