# OpenReview forum: "Concentration inequalities under sub-Gaussian and sub-exponential conditions"
_NeurIPS.cc/2021/Conference — NeurIPS 2021 Poster_

### Official Review · Reviewer_CLFV · 2021-07-05

**Rating:** 6
**Confidence:** 3

**Summary:**

The paper provides sub-Gaussian and sub-exponential concentration bounds for functions of independent variables and applies these bounds to Principal Components Analysis (PCA) and linear regression.

**Ethical Concerns:**

I am not aware of ethical concerns regarding this paper

**Limitations And Societal Impact:**

The authors discussed a limitation of their work in the conclusion

**Main Review:**

The paper studies random variables of the form $f(X)$,  where $f$ is a function of $n$ variables and $X$ is  a vector of $n$ independent random variables. The paper defines the $k$-centered conditional version $f_k$ of $f(X)$, for $1\le k\le n$,  and proves concentration bounds on $f(X)$ in terms of sub-Gaussian and sub-exponential norms associated with the $f_k$'s. These bounds are interesting. For instance, Theorem 1 generalizes existing concentration bounds on the sum of independent random variables.  On the negative side, I think that the machine-learning applications of these techniques are not sufficiently developped to attract the audience of NIPS. Also, the statement of Lemma 11 is erronous: the inequality in Lemma 11 (i) does not holds if $Y$ is a negative constant. Here are other comments:

--The statement in lines 124--126 should be clarified.

--Corollary 9. Something is missing, because the inequality does not hold with probability 1

--Equation after line 217: typo in independet.

--Last equation in Theorem 10. I think $\sum_k$ should be replaced by $\sum_i$

--Lemma 11: variables -> variable.

--After feedback, I have increased my score.

**Time Spent Reviewing:**

8

---

> ### Author Response · Authors · 2021-08-08
> **Many thanks for your comments and suggestions**
>
> Many thanks for your comments and suggestions.
>
> Apart form the significance of our work to machine learning theory, we agree on all the other specific points you raise and promise fixes.
>
> Special thanks for mentioning the error in Lemma 11. It should read "for any centered random variable", and the first sentence in the proof should be deleted.

---

> ### Comment · Area_Chair_Tkyf · 2021-08-08
> **Corollary 9**
>
> Reviewer CLFV wrote: "Corollary 9. Something is missing, because the inequality does not hold with probability 1". Has this comment been addressed? Is it fixable?

---

> > ### Author Response · Authors · 2021-08-08
> > **It is fixable**
> >
> > We forgot to write "with probability at least $ 1-\delta $ it holds that ...", a phrase that appears in several other statements, where the confidence parameter $\delta$ appears, but in this place we forgot to write it.

---

### Official Review · Reviewer_tbKF · 2021-07-14

**Rating:** 7
**Confidence:** 3

**Summary:**

McDiarmid's inequality is one of the workhorses of modern machine learning. In its simplest form, however, it requires the function under consideration to verify a bounded differences property. This work (as others before it) investigates extensions of this idea beyond this stringent requirement.  Using the entropy method [Boucheron et al., 2013, Section 6],  the authors prove concentration inequalities for functions of independent random variables for which the *conditional versions* (random function that keeps all arguments but one fixed) enjoy sub-Gaussianity/sub-exponentiality (instead of being bounded as for *vanilla McDiarmid*).  They further put the new tools in application for vector valued concentration, principal subspace analysis and generalization through Rademacher complexity and algorithmic stability.

**Ethical Concerns:**

None.

**Limitations And Societal Impact:**

Adequately addressed.

**Main Review:**

To the best of this reviewer's knowledge, Theorem 2 and 3 are novel.
These newly developed tools would be of value to the machine learning community, as exemplified by the authors.
This reviewer recommends acceptance, modulo a few clarifications and fixes (detailed below).

* It is not clear to this reviewer why the authors decided to redefine the sub-Gaussian and sub-exponential  norms  in  a  way  that  they  say  is  different  from  Vershynin  [2018].   For  previously  defined norms, centering properties in Lemma 4 are well-known [Vershynin, 2018, Lemma 2.6.8, Exercise 2.7.10].  If this alternative definition has been given before,  please add a reference.  If it is new,  please provide some motivation.

* Minor detail:
    * p.1  The current title is uninformative. This reviewer encourages the authors to revisit it.
    * p.2 Recall briefly definitions of norms on random variables (usual $L_p$ norms in terms of expectations).
    * L.58  ”does not dependent”
    * L.237 $\phi_2 \to \psi_2$

**References**

* S. Boucheron,  G. Lugosi,  and P. Massart. Concentration inequalities: A non asymptotic theory of independence. Oxford university press, 2013.

* R. Vershynin. High-dimensional probability: An introduction with applications in data science, volume 47.  Cambridge university press, 2018.

**Time Spent Reviewing:**

3

---

> ### Author Response · Authors · 2021-08-08
> **Many thanks for your comments and suggestions.**
>
> Many thanks for your comments and suggestions.
>
> Redefinition of norms: that our definitions are equivalent to the customary ones is the content of Proposition 2.7.1 and Proposition 2.5.2 in Vershynin's book. We give this\ citation, but we plan to clarify this point further in the next version. The multiplicative constants with which the equivalent norms are related, lead to different constants for the centering properties (Lemma 4 (ii) in our paper). The purpose of the redefinition was to simplify the power-series argument in the proofs of the concentration inequalities (Lemma 12 and Lemma 14).
>
> We agree on the other specific points you raise and promise fixes. As regards the title, reviewer hVqo also dislikes it and we begin to dislike it ourselves, but it is unclear if we can still change it, should the paper be accepted. An alternative title could be "Concentration inequalities under sub-Gaussian and sub-exponential conditions".

---

> > ### Comment · Reviewer_tbKF · 2021-08-17
> > **Changed score**
> >
> > The reviewer thanks the authors for their response, and changes the score accordingly from 6 to 7: Accept.

---

### Official Review · Reviewer_SxeE · 2021-07-14

**Rating:** 9
**Confidence:** 3

**Summary:**

This paper derives concentration inequalities for functions of independent random variables that are sub-Gaussian or sub-exponential. Applications to vector-valued concentration, PCA, and Rademacher generalization bounds are provided.

**Main Review:**

The main contributions are the sub-exponential bounds, and their applications are compelling. I think the results are highly relevant. My main suggestions are in the presentation of the material.

I would suggest the following reorganization. You could include the notation as a small piece of the introduction section. Since there are quite a few preliminaries, you could insert a “Preliminaries” section between the results and applications, in which you give the facts that are currently in the notation section along with the material from what is now Section 4.1.

Writing is generally polished, though I found some grammar/style issues.
- “circumnavigated”→ circumvented
- PCA is more standard than PSA (in any case be consistent with the abstract)
- Something wrong with third sentence of the introduction
- f_k(X)(x) is heavy notation. Please avoid double parentheses.
- line 16: In situations, where → no comma
- line 232: missing theorem environment


**Time Spent Reviewing:**

1

---

> ### Author Response · Authors · 2021-08-08
> **Many thanks for your comments and suggestions**
>
> Many thanks for your comments and suggestions.
>
> We plan to move Section 4.1 into Section 2, which would become the
> "Preliminaries"-section you mention.
>
> We agree on all the specific points you raise and promise fixes.

---

### Official Review · Reviewer_hVqo · 2021-07-15

**Rating:** 7
**Confidence:** 4

**Summary:**

This paper shows generalizations of the method of bounded differences to a broad class of functions with sub-Gaussian and sub-exponential behavior.
Important applications include Rademacher averages, and bounding the generalization error of PCA. The paper is quite technical, but the machinery developed therein is highly general, and seems to this reviewer to be of great interest in a broad variety of machine learning settings.


**Limitations And Societal Impact:**

Yes.  societal impact: largely N/A

**Main Review:**

Overall, this is a very nice paper, with strong results that are less restrictive than prior work.
This is a well-trodden area, but this paper makes significant contributions driving bounds that are both more straightforward to use, and, in many cases, sharper.  Having read many of the cited works, coming away with little concept of how to apply them to adapt standard learning methodology to unbounded functional families, several of the results here were quite refreshing, and did seem directly applicable to important learning settings. That being said, there are some connections to work in uniform convergence, that are now quite classic, that are missing.  There is also, of course, some work left to do here, which could be more clearly outlined in this paper.  This reviewer holds that the contribution already is sufficient for a strong NeurIPS publication.

The comment on line 178 about the classical approach is quite misleading; While Bartlett (2002) gives this approach, Talagrand’s inequality for the supremum of empirical processes was well known by the time the Rademacher average entered the popular machine learning dialogue, and Bousquet’s inequality was proved around this same time (See [1], chapter 12).
These bounds depend on ranges substantially less than the corresponding McDiarmid bounds. What would be the advantage of applying your method theorem 8 over the Bousquet bound, with a correction for random variables escaping a bounded set with negligible probability?  Alternatively, could your results be used to derive an analogue of the Bousquet inequality relaxing the boundedness assumption?


One nitpick: the title is rather strange to me. What is a Hoeffding-type bound, if not one that shows sub-Gaussian guarantees under a bounded range assumption? If we take away the bounded range assumption, as this paper so valiantly accomplishes, I would hardly call the result Hoeffding-type.


Specific Points:

The explanation on line 52 about the function being function valued was very confusing; I understood it, but there must be a better way to express this.
Should there be an X’ involved in the expectation in the display math after line 57?

line 60: the order of the norms does not match their order in expression 1.

line 61: are the usual LP norms the pth-root of the p-moments? I suspect these definitions are quite familiar to readers of Vershynin, but an explanation in terms of moments would be beneficial for some audiences.  Furthermore section 4.1 would be much more helpful if the information were presented here.

Line 75: “A similar results to” -> “A similar result to”

Line 75: The Kontorovich work uses a symmetric notion of sub-Gaussian diameter; I believe this result is significantly more general, so it reads to me like you're under-selling your work here.

76: The statement about essential supremum reads poorly, I would prefer to see this in algebraic notation.

Lines 81 to 83 (and in general much in the paper): I believe this can be expressed in the modern parlance as sub-gamma random variables; in particular this is helpful, as you draw a large parallel to the (sub-gamma) Bernstein inequality. It also raises the question of whether similar bounds can be shown for sub-Poisson random variables, as has often been the case historically, e.g., Bernstein to Bennett, and Talagrand to Bousquet inequalities.

Some dressings on lemma 5 would be appreciated.  Clearly Hoeffding’s lemma handles the sub-Gaussian case, and it appears you bound moments and the sub-exponential constants under similar assumptions?

line 135: Concentration of norms sounds like the expectation of the norm, whereas you bound the norm of the expectation.
Furthermore, defining the norm of the X as simply ||.|| is rather confusing, as there are a lot of different norms and vertical double bars already. Perhaps writing as ||.||_X would be easier on the reader?

Theorem 7: Is it possible to get an absolute value inside of the supremum with a symmetry argument (at no additional cost)? Furthermore, is it possible to remove the union bound by working directly with the MGFs before applying Markov’s inequality? The result is nice, but it does not seem necessary to have $log(2 / \delta)$ on this 1-tail bound.


It may be helpful to remind readers that the random variable defined on 170 is generally termed the supremum deviation.

176: I would like to see an example of this. I guess I can see how, e.g., the Massart and Khintchine inequalities can bound empirical Rademacher averages in terms of only empirical (raw) variances, but how does one translate this into a bound on expected Rademacher averages? Even knowing the maximum raw variance (wimpy) seems insufficient; it seems one would need to know the weak variance, but the relationship between weak and wimpy variance relies on boundedness in the contraction inequality application.


The quantity described as the entropy on line 228 is, of course the entropy functional used in e.g., Ledoux's entropy method, however I would read “entropy of a real valued random variable” as “Shannon differential entropy” if it were not further qualified.
References seem to be to Maurer 2012, whereas perhaps citing Ledoux would be more appropriate.

Must the definition of tilted expectation be written in this manner, or must it be defined at all, if only used once?  Many authors use subscripts on expectations to denote the random variable that the expectation is being taken with respect to.



[1] “Concentration inequalities: A nonasymptotic theory of independence,” Stéphane Boucheron, Gábor Lugosi, et Pascal Massart


**Time Spent Reviewing:**

12

---

> ### Author Response · Authors · 2021-08-08
> **Many thanks for your comments and suggestions**
>
> Many thanks for your comments and suggestions.
>
> Comment on l. 178 misleading: with "classical approach" we meant the standard application to machine learning, not the already established use for empirical processes. Perhaps "standard approach"  would have been better.
>
> Using Bousquet's inequality instead: we are unclear how the correction you mention could be implemented. Finding an analogue of Bousquet's inequality without boundedness assumption seems an interesting, but also very challenging problem.
>
> About the title: reviewer tbKF also complained about the title. We are beginning to dislike it ourselves, but it is unclear if we can still change it, should the paper be accepted. An alternative title could be "Concentration inequalities under sub-Gaussian and sub-exponential conditions".
>
> We agree on most of the specific points you raise and promise fixes wherever possible. Three remarks:
>
> --  l. 61: We plan to move Section 4.1 into Section 2.
>
> -- l. 176: An example would be $\mathcal{H}=\{ x\mapsto \langle
> w,x\rangle :\Vert w\Vert \leq 1\} $ with $X_{i}$
> unbounded in a Hilbert space, but ${E}[\Vert
> X_{i}\Vert^{2}] \leq 1$. Then the $h(X_i)$ are
> not bounded variables, but the orthonormality of the Rademacher variables
> and the bound on ${E}\left[ \left\Vert X_{i}\right\Vert ^{2}\right] $
> would make the expected Rademacher average $\leq 2/\sqrt{n}$.
>
> -- The entropy in l. 228 is related to Ledoux's entropy (on p 91 in his book) as $S(Y) = $ Ent$_\mu(e^{Y})/{E}[e^{Y}]$.

---

> > ### Comment · Reviewer_hVqo · 2021-08-13
> > **Thank You**
> >
> >
> > Thank you, this is a good example. I did think of Hilbert spaces, but I was stumped because I was trying to think of something that could be estimated based on the sample, but I see now that you need to make an assumption about the distribution, or at least about the extreme tails. I would recommend adding this example to the paper if possible.  On that note, since Proposition 6 bounds the expected norm, could this be used in conjunction with this result to derive a bound on Rademacher averages?

---

> > > ### Author Response · Authors · 2021-08-13
> > > **Expected Rademacher averages**
> > >
> > > Thank you for your comment. In a sense we do have this example already in the paper: in the proof of Corollary 9, which is less visible, because it is given in the supplement. Also it is more specialized than the above.
> > > The answer to your question is yes, because the 2-norm is bounded by twice the subexponential norm, and this is also what we use in the proof of Corollary 9.

---

### Decision · Program_Chairs · 2021-09-27

**Decision:**

Accept (Poster)

**Comment:**

The referees are in agreement that this submission provides novel concentration-of-measure inequalities. It is very much within the conference scope and of sufficient interest and novelty. All of the referee objections have been addressed during the discussion phase.